# Improving Consistency for Text Summarization with Energy Functions

**Qi Zeng[1], Qingyu Yin[2], Zheng Li[2], Yifan Gao[2], Sreyashi Nag[2], Zhengyang Wang[2],**
**Bing Yin[2], Heng Ji[2], Chao Zhang[2]**
[1]University of Illinois Urbana-Champaign [2]Amazon
[1]qizeng2@illinois.edu
[2]{qingyy, amzzhe, yifangao, sreyanag, zhengywa,
alexbyin, jihj, zhanpcha}@amazon.com

## Abstract

Current abstractive summarization models often generate inconsistent content, i.e. texts that are not directly inferable from the source document, are not consistent with respect to world knowledge, or are self-contradictory. These inconsistencies motivate a new consistency taxonomy that we define as faithfulness, factuality, and self-supportiveness. However, most recent work on reducing inconsistency in document summarization only focuses on faithfulness detection and correction while ignoring other inconsistency phenomena, which limits the model's scalability. To improve the general consistency we introduce EnergySum, where we apply the Residual Energy-based Model by designing energy scorers that reflect each type of consistency. These energy scores are utilized in candidate re-ranking during the sampling process. Experiments on XSUM and CNN/DM datasets show that EnergySum mitigates the trade-off between accuracy and consistency.

## 1 Introduction

While performing well in terms of overlap-based metrics like ROUGE (Lin, 2004) and BERTScore (Zhang et al., 2020), current abstractive summarization methods often generate inconsistent content due to the inherently noisy dataset and the discrepancy between maximum likelihood estimation based training objectives and consistency measurements. Inconsistency content in abstractive summarization has different interpretations, including text that is not directly inferable from the source document, is not factual with respect to world knowledge and commonsense, or is self-contradictory. We formalize the categorization of consistency into **faithfulness, factuality, and self-supportiveness**. Table 1 illustrates different types of consistency errors.

Most previous methods improve consistency in document summarization by filtering out noisy training samples (Kang and Hashimoto, 2020), applying contrastive learning (Cao and Wang, 2021), post-editing (Cao et al., 2020), etc., with a limited scope of consistency to faithfulness. However, addressing inconsistency solely in terms of faithfulness is inadequate. Unlike extractive methods, abstractive summarization introduces new content into the summary that is not directly copied from the source document and is not necessarily irrelevant. Hence, detecting and alleviating inconsistency calls for the introduction of a larger reference corpus alongside the source document. Factuality compares the generated content against world knowledge, while self-supportiveness verifies whether the generated sentence is consistent with its preceding one.

In addition, consistency is measured on the entire prediction sequence while existing summarization objectives evaluate conditional distributions for individual tokens and lack global control over predictions. These motivate us to apply the Residual Energy-based Model (REBM) (Deng et al., 2020) framework to document summarization, which jointly trains a summarizer and a discriminator that learns to assign high scores to consistent summaries and low scores to inconsistent ones. The advantage of the energy-based methods (He et al., 2021) is that they score the entire input simultaneously and avoid local normalization traps, offering a natural solution to address this issue.

Therefore, we introduce **EnergySum** that adapts the REBM framework for improving consistency. We design the energy functions that reflect each type of consistency and are agnostic to summarization model instances. We propose joint inference where energy scorers cooperate with decoding searching strategies in the candidate re-ranking step. In summary, our contributions include:

- We formalize the categorization of consistency in document summarization into faith-

| Source document |
| --- |
| *Oscar-winning actress Angelina Jolie* is visiting *Iraq* to boost what she sees as lagging efforts to deal with the problems of 2 million "very very vulnerable" internally displaced people in the wartorn country... *More than 4.2 million* Iraqis have fled their homes, around 2 million to neighboring states, mostly Syria and Jordan... |

| Consistency type | Example summary |
| --- | --- |
| **Faithfulness:** The text is directly inferable from the source document. | ... More than 5 million Iraqis have fled homes, 2 million to neighboring states ... |
| **Factuality:** The text contains hallucinated but true content referring to world knowledge. | American actress Angelina Jolie visits Iraq to boost efforts to help internally displaced refugees... |
| **Self-supportiveness:** The text does not contain self-contradictory errors. | ... 2 million Iraqis have fled to neighboring states. Another 2 million are displaced domestically inside Syria and Jordan... |

Table 1: Example summaries with different types of inconsistency. The errors in the sample summaries are in red.

fulness, factuality, and self-supportiveness.

- We propose the EnergySum framework, which includes consistency-constrained energy scorers and joint inference. We are the first to introduce energy-based methods to consistent document summarization.
- We conduct experiments on XSUM and CNN/DM datasets to validate the effectiveness of EnergySum.

## 2 Related Work

Recent work in consistent abstractive summarization has been looking into reducing entity-based hallucinations. Nie et al. (2019) reduce hallucinations by integrating a language understanding module for data refinement with self-training iterations. Zhao et al. (2020) reduce quantity hallucination by verifying quantity entities and promoting less hallucinated summaries. Kang and Hashimoto (2020) propose a loss truncation training algorithm that filters out noisy training samples which may lead to hallucination. Cao et al. (2022) detect factual hallucinations by utilizing the entity's prior and posterior probabilities according to the pretrained and fine-tuned masked language models and use it as a reward signal in reinforcement learning. Dixit et al. (2023) propose a candidate summary re-ranking technique for contrastive summarization training to improve both faithfulness and summary quality. Zhang et al. (2023) use Information Extraction (IE) in a multi-task training manner to improve factual consistency of multi-document summarization.

The most related work to ours is CLIFF (Cao and Wang, 2021), which applies contrastive learning to abstractive summarization by designing negative sample generation strategies to resemble errors made commonly by state-of-the-art summarization models. Though both are training discriminators

on top of decoders with NCE loss, our work differs in the structure of discriminators, the training loss, and the inference process.

Correction-based methods are proposed for mitigating the trade-off between consistency improvement and ROUGE-based accuracy measurement decrease. Cao et al. (2020) propose a post-editing corrector module trained on synthetic examples, where heuristic transformations are inspired by an error analysis on reference summaries. Span-Fact (Dong et al., 2020) is a factual correction model that leverages knowledge learned from Question Answering models to make corrections in system-generated summaries via span selection. Zhu et al. (2021) propose a fact-aware summarization model to integrate factual relations into the summary generation process and a factual corrector model in the form of a finetuned denoising auto-encoder.

Automatic consistency evaluation models can be roughly classified into entailment-based and QA-based methods. Entailment-based metrics (Kryscinski et al., 2020; Laban et al., 2022; Ribeiro et al., 2022) train classification models to predict if the summary is entailed by the source document. Meanwhile, QA-based metrics (Fabbri et al., 2022; Scialom et al., 2021; Durmus et al., 2020) generate questions based on the input summary and document, then apply QA models to answer the questions and compare the answers to calculate a faithfulness score. Chan et al. (2023) propose a multi-label classification model grounded on semantic role labeling to predict the types of faithfulness error in a summary. Ladhak et al. (2022) evaluate effective faithfulness of summarization systems with a faithfulness-abstractiveness trade-off curve. Cheang et al. (2023) evaluate and analyze the faithfulness of pre-trained summarization models on dynamically evolving data.

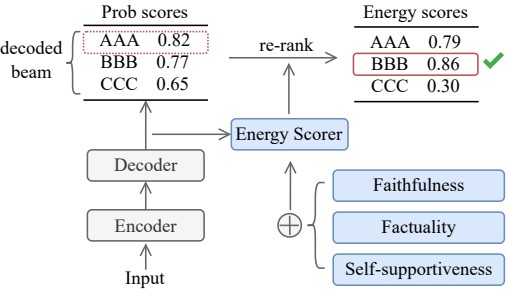

Figure 1: Overview of EnergySum framework. The energy scorer is a discriminator consisting of three consistency-constrained energy functions. During inference, we re-rank the decoded beam of summaries by energy scores.

# 3 Method

In the proposed EnergySum framework, we design energy scorers that correlate each type of consistency and integrate energy scores in candidate re-ranking during sampling.

## 3.1 Background: Energy-Based Models

Energy-Based Model (EBM) (LeCun et al., 2006) is a general learning framework that assigns an un-normalized energy score to any given input. EBM has been applied in machine translation to solve the discrepancy between the training objective (Maximum Likelihood Estimation) and the task measure (BLEU) (Bhattacharyya et al., 2021), and in improving calibration in natural language understanding (He et al., 2021).

Residual Energy-Based Models (R-EBMs) (Deng et al., 2020) are introduced to text generation, which use EBM to learn from the residual errors of an auto-regressive generator to reduce the gap between the model and data distributions: $P_\theta \propto P_{LM}(x)\exp(-E_\theta(x))$, where $P_{LM}$ is a locally normalized language model and $E_\theta$ is the energy function. Li et al. (2021) further applies R-EMBs to end-to-end speech recognition.

Energy functions have also been used as constraints in text generation. The COLD decoding framework (Qin et al., 2022) unifies constrained generation by specifying constraints through an energy function, then performing efficient differentiable reasoning over the constraints through gradient-based sampling.

## 3.2 Energy Functions for Consistency

Energy functions solve the discrepancy between MLE-based training objectives and consistency measurements. General-purpose energy function designs are usually as simple as the mean pooling over the last encoder/decoder layer logits. To improve consistency, we propose three energy functions and use their weighted sum as the final energy function in the Noise Contrastive Estimation loss.

$$\mathcal{E}(x, y, \hat{y}) = \lambda_1 \mathcal{E}_i(y, \hat{y}) + \lambda_2 \mathcal{E}_i(x, \hat{y}) + \lambda_3 \mathcal{E}_i(\hat{y})$$

where $x$ is the input document, $y$ is the reference summary, and $\hat{y}$ is the generated summary.

**Faithfulness.** Following Qin et al. (2022) we use EISL (Edit-Invariant Sequence Loss) (Liu et al., 2022) as a similarity measure. This n-gram matching function can be seen as a differentiable approximation to the BLEU-n metric. Its computation is essentially a convolution operation on the candidate sequences using target n-grams as kernels.

$$\mathcal{E}_1(y, \hat{y}) = \text{EISL}(y, \hat{y})$$

During training, we use the reference summary to measure faithfulness for stable and efficient training. However, it cannot avoid dataset noise from annotation as it is based on the assumption that the reference summary is correct. Also, the gold summary is not available during inference.

**Factuality**. Cao et al. (2022) propose to detect factual hallucinations by utilizing the entity's prior and posterior probabilities according to the pre-trained and fine-tuned masked language models as classifier inputs. It is still under exploration how these two distributions work together for factual hallucinations. To apply this measure, we first initiate and freeze the pretrained BARTlarge model as the prior model. A classifier $\gamma$ takes the concatenation of outputs from the prior and posterior models as its input.

$$\mathcal{E}_2(x, \hat{y}) = \gamma(p_{prior}(\hat{y}|x), p_{posterior}(\hat{y}|x))$$

**Self-supportiveness**. A non-linear layer $\phi$ on top of the decoder outputs detects self-supportiveness in the generated summary.

$$\mathcal{E}_3(\hat{y}) = \phi(p(\hat{y}))$$

| Dataset | Model | ROUGE-1 | ROUGE-2 | ROUGE-L | BERTSCORE | FEQA | ENTFA | DAESS |
|---|---|---|---|---|---|---|---|---|
| XSUM | Human | - | - | - | - | 18.95 | 72.27 | - |
| | BARTlarge | 43.64 | 20.04 | 34.34 | 91.56 | 29.13 | 68.38 | - |
| | FASUM | 30.61 | 10.06 | 23.97 | 88.53 | 18.38 | 55.83 | - |
| | FASUM+FC | 30.53 | 10.00 | 23.89 | 88.58 | 19.77 | 54.91 | - |
| | Losstrunc | 41.73 | 17.88 | 32.68 | 91.24 | 28.94 | 66.31 | - |
| | CLIFF | **42.07** | **18.50** | 32.82 | 91.29 | 25.28 | **83.87** | - |
| | **EnergySum** | 41.69 | 18.12 | **32.98** | **91.44** | **30.26** | 68.45 | - |
| CNN/DM | Human | - | - | - | - | 30.94 | 91.46 | 99.95 |
| | BARTlarge | 43.86 | 21.07 | 40.74 | 88.70 | 18.06 | 63.50 | 99.92 |
| | FASUM | 40.83 | 17.94 | 37.78 | 88.08 | 18.75 | 61.23 | **99.89** |
| | FASUM+FC | 40.68 | 17.77 | 37.63 | 88.24 | 18.74 | 60.53 | **99.89** |
| | Losstrunc | 36.37 | 17.35 | 34.21 | 87.72 | 11.58 | 65.90 | 99.65 |
| | CLIFF | 42.15 | 19.82 | 38.91 | 87.95 | 21.33 | 64.90 | 99.86 |
| | **EnergySum** | **43.38** | **20.45** | **40.27** | **88.27** | **41.92** | 66.43 | 99.89 |

Table 2: Results(%) on XSUM and CNN/DM datasets. ROUGE and BERTSCORE indicate accuracy. FEQA, ENTFA, and DAESS evaluate faithfulness, factuality, and self-supportiveness, respectively. For all scores, the higher the better.

### 3.3 Training Loss

The pretrained language model is fine-tuned using the cross entropy loss $\mathcal{L}_{CE}$:

$$\mathcal{L}_{CE} = -\sum y_i \log \hat{y}_i$$

For stable and effective training of the discriminator, we combine the two squared hinge loss $\mathcal{L}_{\mathcal{E}}$ (Liu et al., 2020) and the similarity-based NCE loss $\mathcal{L}_{\text{sim}}$ (Cao and Wang, 2021).

$$\mathcal{L}_{\mathcal{E}} = \mathbb{E}_{x_+}(\max(0, \hat{\mathcal{E}}_\theta(x_+)) - m_1))^2 \\ + \mathbb{E}_{x_-}(\max(0, m_2 - \hat{\mathcal{E}}_\theta(x_+)))^2 \quad (1)$$

$m_1$ and $m_2$ are margin hyper-parameters with which the loss function penalizes samples with energy $\hat{\mathcal{E}} \in [m_1, m2]$.

$$\mathcal{L}_{\text{sim}} = -\mathbb{E} \log \frac{\exp(\text{sim}(h_i, h_j))}{\sum \exp(\text{sim}(h_i, h_k))}$$

In the above loss, $P$ and $N$ are the positive sample set and the negative sample set, $y_i, y_j \in P, y_j \neq y_i, y_k \in P \cup N, y_k \neq y_i$. $h_i, h_j, h_k$ are representations for summaries $y_i, y_j, y_k$, and $\text{sim}(\cdot, \cdot)$ calculates the cosine similarity between summary representation.

The final training loss is a combination of the above losses:

$$\mathcal{L} = \mathcal{L}_{CE} + \lambda_{\mathcal{E}}\mathcal{L}_{\mathcal{E}} + \lambda_{\text{sim}}\mathcal{L}_{\text{sim}}$$

### 3.4 Joint Inference

Previous work (Deng et al., 2020) suggests that a sample-resample procedure is similar to exact sampling from the joint distribution. Therefore, we modify the sampling process by inserting the energy scores into the candidate re-ranking step.

In decoding, a batch of sentence candidates is generated and scored for each input. We replace the generation probability scores with energy scores for the candidates and re-rank the batch. Since beam search is more likely to generate similar results, where re-ranking takes less effect, we select diverse beam search (Vijayakumar et al., 2016) as the default searching strategy.

## 4 Experiments

### 4.1 Setup

**Datasets and Baselines**. We compare our method with BARTlarge (Lewis et al., 2020), LOSSTRUNC (Kang and Hashimoto, 2020), FASUM and its variant FASUM+FC (Zhu et al., 2021) and CLIFF (Cao and Wang, 2021) on XSUM (Narayan et al., 2018) and CNN/DM (Nallapati et al., 2016) datasets. Human baseline refers to the human-written reference summaries.

**Evaluation Metrics**. We evaluate accuracy with ROUGE (Lin and Hovy, 2003) and BERTScore (Zhang et al., 2020). For faithfulness and factuality, we measure with FEQA (Durmus et al., 2020) and ENTFA (Cao et al., 2022), respectively. Since there is no existing metric for self-supportiveness, we propose DAESS, which splits the multi-sentence summary and adapts DAE (Goyal and Durrett, 2021) to compare every pair of sentences in one summary. The summaries in the XSUM dataset are usually one sentence, so we only evaluate DAESS on the CNN/DM dataset.

**Implementation Details**. We instantiate EnergySum and Losstrunc both with the pretrained BARTlarge model[1]. The margin hyperparameters $m_1 = -10, m_2 = -5$ in $\mathcal{L}_{\mathcal{E}}$ are selected by performance on the development set.

For FASUM, we evaluate the provided prediction files as the code is not publicly available. Note that their provided test set file is slightly different than the standard test set split. For all other experiments, each model is trained for $15000$ steps, the learning rate is set to $1e-3$, the max token in one batch is set to $4096$, the update frequency is $16$, and the optimizer is Adam with $500$ warm up steps. The hyperparameter $c$ in Losstrunc is set to $0.3$.

For numerical consistency, all experiment results are averaged across three random runs. On average it takes approximately ten hours to train a model with one Tesla A100 GPU with 40GB DRAM. Since evaluating FEQA over the whole test set is time costly, we randomly sample 500 document-summary pairs to calculate the scores.

### 4.2 Results and Discussion

Table 2 shows that EnergySum improves faithfulness with comparable accuracy performance on both XSUM and CNN/DM compared to BARTlarge. All consistency improvement baselines have lower overlapped-based accuracy than BARTlarge, showing the trade-off between MLE-based training and consistency training. Nevertheless, our method hurts less from such a trade-off and still has comparable accuracy performance.

Human-written gold summaries usually represent the upper bound of the performance. However, the human baseline has lower FEQA (faithfulness) performance, indicating the existence of noise in the dataset. Self-supportiveness scores are all close to 100%, implying that self-supportiveness is not a challenging problem for current summarization systems and also calling for a more fine-grained evaluation metric.

There is also a trade-off between the sampling method selection and the overall performance. Joint inference can only be applied to searching strategies where the searched candidates are diverse, which in general performs worse than regular beam search.

---

[1] https://github.com/facebookresearch/fairseq/tree/main/examples/bart

## 5 Conclusion

We propose to apply the Residual EBM framework with energy scorers and joint inference to improve consistency in document summarization. Experiments on XSUM and CNN/DM datasets show that EnergySum mitigates the trade-off between accuracy and consistency. Direct extensions of this work include proposing more fine-grained data augmentation strategies and investigating the relation between prediction certainty and energy scores.

## Limitations

This work on consistent document summarization has limitations in terms of data scope and task configuration. First, EnergySum learns from common errors simulated by data augmentation strategies, which could limit its application in more diverse contexts. Second, EnergySum predicts sentence-level scores and thus cannot detect span-level errors or predict error types.

## Ethics Statement

The summaries generated by our model may still contain hallucinations, which may lead to misunderstandings of the original documents. The XSUM and CNN/DM datasets used in this study mainly focus on the news domain, which might introduce biases when applied to documents in other domains.

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
