# OpenReview forum: "Improving Consistency for Text Summarization with Energy Functions"
_EMNLP/2023/Conference — EMNLP 2023 Findings_

### Official Review · Reviewer_2Tfi · 2023-07-22

**Typos Grammar Style And Presentation Improvements:** The text looks fine.
**Soundness:** 4

**Excitement:**

3: Ambivalent: It has merits (e.g., it reports state-of-the-art results, the idea is nice), but there are key weaknesses (e.g., it describes incremental work), and it can significantly benefit from another round of revision. However, I won't object to accepting it if my co-reviewers champion it.

**Missing References:**

Momentum Calibration for Text Generation
Xingxing Zhang, Yiran Liu, Xun Wang, Pengcheng He, Yang Yu, Si-Qing Chen, Wayne Xiong, Furu Wei

**Paper Topic And Main Contributions:**

The paper suggests a new method for abstractive text summarization that is focused on factual correctness and faithfulness. The method uses energy scorers that reflect the three types of consistency; these scores are used
in candidate re-ranking.

**Questions For The Authors:**

l.050 Some facts in a document can be unrelated to world knowledge, such as 'She ate an apple'.

Table 1: there is some mix-up with examples here. The 1st example is a hallucination and the second is faithful.

Table 3, CNN/DM results: The MoCa model reports R1 of over 0.48, so the results of competing models do not reflect state-of-the-art. Please survey this paper and address the results in it:

Momentum Calibration for Text Generation
Xingxing Zhang, Yiran Liu, Xun Wang, Pengcheng He, Yang Yu, Si-Qing Chen, Wayne Xiong, Furu Wei

Table 2, especially part 2: are the differences from the next best score statistically significant?

l.230-231 Why? Are there papers that prove this statement?

l.242-243 This is a non-standard notation. Human generated summary are usually called 'gold' or 'ground truth', and Oracle refers to an automated system that performs global optimization (e.g., sentence selection) relying on the known ground truth.

**Reasons To Accept:**

The paper is well-written, and the results are described in a clear way. The ROUGE scores are good but do not reach the current state-of-the-art.

**Reasons To Reject:**

The ROUGE scores are good but do not reach the current state-of-the-art.

**Reproducibility:**

4: Could mostly reproduce the results, but there may be some variation because of sample variance or minor variations in their interpretation of the protocol or method.

**Reviewer Confidence:**

4: Quite sure. I tried to check the important points carefully. It's unlikely, though conceivable, that I missed something that should affect my ratings.

---

> ### Author Rebuttal · Authors · 2023-08-28
>
> 1. Table 1: There is some mix-up with examples here. The 1st example is a hallucination and the second is faithful.
>
> The first example is unfaithful because “More than 5 million” contradicts “More than 4.2 million” in the document. The second example is factual hallucination because Angelina Jolie is indeed an American actress but this fact is not mentioned in the document.
>
> We will add clarification in the figure caption to improve clarity.
>
> 2. Table 3, CNN/DM results: The MoCa model reports R1 of over 0.48, so the results of competing models do not reflect state-of-the-art. Please survey this paper and address the results in it: Momentum Calibration for Text Generation Xingxing Zhang, Yiran Liu, Xun Wang, Pengcheng He, Yang Yu, Si-Qing Chen, Wayne Xiong, Furu Wei
>
> Thank you for your suggestion!
>
> 3. l.230-231 Why? Are there papers that prove this statement?
>
> Thank you for raising this question. We will add the citations.
>
> It has long been understood that solutions decoded by Beam Search are lacking in diversity. Please refer to the following papers [1-5]. As [1] claims “BS explores the search space in a greedy left-right fashion retaining only the top-B candidates - resulting in sequences that differ only slightly from each other. ” The reasons why beam search is lack of diversity can also be found in [1].
>
> [1] Vijayakumar, A.K., Cogswell, M., Selvaraju, R.R., Sun, Q., Lee, S., Crandall, D. and Batra, D., 2016. Diverse beam search: Decoding diverse solutions from neural sequence models. arXiv preprint arXiv:1610.02424.
> [2] Finkel, J.R., Manning, C.D. and Ng, A.Y., 2006, July. Solving the problem of cascading errors: Approximate bayesian inference for linguistic annotation pipelines. In Proceedings of the 2006 Conference on Empirical Methods in Natural Language Processing (pp. 618-626).
> [3] Gimpel, K., Batra, D., Dyer, C. and Shakhnarovich, G., 2013, October. A systematic exploration of diversity in machine translation. In Proceedings of the 2013 Conference on Empirical Methods in Natural Language Processing (pp. 1100-1111).
> [4] ​​Li, J., Galley, M., Brockett, C., Gao, J. and Dolan, B., 2015. A diversity-promoting objective function for neural conversation models. arXiv preprint arXiv:1510.03055.
> [5] Li, J. and Jurafsky, D., 2016. Mutual information and diverse decoding improve neural machine translation. arXiv preprint arXiv:1601.00372.
>
>
>
> 4. l.242-243 This is a non-standard notation. Human generated summary are usually called 'gold' or 'ground truth', and Oracle refers to an automated system that performs global optimization (e.g., sentence selection) relying on the known ground truth.
>
> Thank you for your suggestion. We will correct the usage of this notation.

---

### Official Review · Reviewer_zq8F · 2023-08-03

**Soundness:** 2

**Excitement:**

3: Ambivalent: It has merits (e.g., it reports state-of-the-art results, the idea is nice), but there are key weaknesses (e.g., it describes incremental work), and it can significantly benefit from another round of revision. However, I won't object to accepting it if my co-reviewers champion it.

**Missing References:**

* Faisal Ladhak, Esin Durmus, He He, Claire Cardie, and Kathleen McKeown. Faithful or extractive? on mitigating the faithfulness-abstractiveness trade-off in abstractive summarization. ACL 2022
* Tanay Dixit, Fei Wang, and Muhao Chen. Improving Factuality of Abstractive Summarization without Sacrificing Summary Quality. ACL 2023.

**Paper Topic And Main Contributions:**

This paper defines three energy functions to evaluate the faithfulness, factuality, and coherence of summaries. The energy functions are incorporated into both the training and inference process to improve the corresponding properties of system-generated summaries.

**Questions For The Authors:**

* This paper seeks to optimize multiple consistency scores at the same time. Do you observe conflicts among these metrics? Have you considered the trade-off?

**Reasons To Accept:**

* This paper proposes a novel method to integrate consistency-based energy functions into summarization models during training and inference.

**Reasons To Reject:**

* Current results do not show that the proposed method is better than the baselines. The proposed method only outperforms vanilla BART on FEQA and ENTFA among the seven metrics. In addition, automatic evaluation metrics can not provide reliable faithfulness, factuality, and coherence scores, so it's better to conduct human evaluation.
* The proposed method has several components, but no ablation study is provided to show the effect of each component, such as each energy function and using the function during training or inference. Given so many add-on techniques, the performance improvement is too small.
* Some essential method details are missing. The authors only provide a high-level overview of each energy function. It's unclear how to implement these energy functions in practice.

**Reproducibility:**

2: Would be hard pressed to reproduce the results. The contribution depends on data that are simply not available outside the author's institution or consortium; not enough details are provided.

**Reviewer Confidence:**

4: Quite sure. I tried to check the important points carefully. It's unlikely, though conceivable, that I missed something that should affect my ratings.

**Typos Grammar Style And Presentation Improvements:**

* Citations in line 170 and line 183 should not have parentheses.

---

> ### Author Rebuttal · Authors · 2023-08-28
>
> Thank you for your suggestions!
>
> For your question regarding the trade-off between different energy functions, we have observed that they do show interactions and the selection of lamdas will help mitigate the negative effect. We will add clarifications in the next version.
>
> For your concern regarding the performance, as we are comparing SOTA baseline systems with multiple metrics, we achieve SOTA performance in terms of BERT-SCORE, and FEQA on XSUM and CNN/DM, and keep comparable accuracy performance. These facts illustrate the effectiveness of the proposed method.
>
> Regarding your concern about method details, due to space limitations, we cannot provide very detailed information about the implementation. However, we do believe that we have provided the necessary descriptions of the proposed energy function. We will release the code with detailed documentation, which can also help understand the implementation details of these energy functions in practice.

---

### Official Review · Reviewer_qsTq · 2023-08-04

**Soundness:** 3

**Excitement:**

4: Strong: This paper deepens the understanding of some phenomenon or lowers the barriers to an existing research direction.

**Paper Topic And Main Contributions:**

To generate text summaries that are factual, faithful and self-supportive, the paper introduces EnergySum, an energy-based model that uses energy scores associated with the above types of consistency to train a summarization model. During training three different loss functions (energy functions) are introduced to account for the three consistency aspects. At the inference time, the energy scores are computed and the generated sequences are reranked according to these scores. The paper compares the proposed method with similar approaches on two datasets Xsum and CNN/DM and presents the performance of the model.


**Questions For The Authors:**

A. Why is the energy function for faithfulness computed with respect to the reference summary and not the source articles? As also stated by the authors, faithfulness is associated with generating text that is inferable from the source document.

B. What is the intuition behind using the similarity-based NCE loss in equation 1?

C. Some of the parameters in the equations are not described such as P and N.

D. The results show (and the authors emphasize) that self-supportiveness is not a challenging problem for the current systems. Do you think the results would change if you remove the self-supportiveness loss term?

E. Can energy score be applied while decoding with sampling and not beam decoding?

F. Xsum consists of one-line summaries. What role do you think the length of the summaries can play when optimizing for the propose energy functions?


**Reasons To Accept:**

Current LLMs are capable of generating coherent and plausible text however consistency still remains a challenge for these models therefore introducing a model that can capture the different aspects of consistency within these models is an important contribution.

The proposed approach is designed such that it can further be augmented with additional dimensions of consistency (or any other desired metric).

The experimental results show that the proposed approach can outperform the baselines in terms of factuality and faithfulness.


**Reasons To Reject:**

The distinction between the proposed approach and the other relevant summarization works on consistency is not clearly described.

On the experiments side, given that this is a short paper and lack of space, I still believe that more analysis is needed to support the better performance of the model. First, I think the consistency-based metrics still needs human-evaluation on top of the automatic metrics so I guess a human evaluation of the generated summaries on these dimensions could be insightful as well as doing an error analysis to categorize the errors made by system.

Also, ENTFA computes factuality only with respect to the entities therefore it can only cover one aspect of the factuality. I guess considering other factuality metrics as well as the manual evaluation can further emphasize the results in paper 2.

I would also like to see an ablation study of different loss terms and decoding strategies to see how they can affect the results.


**Reproducibility:**

5: Could easily reproduce the results.

**Reviewer Confidence:**

4: Quite sure. I tried to check the important points carefully. It's unlikely, though conceivable, that I missed something that should affect my ratings.

---

> ### Author Rebuttal · Authors · 2023-08-28
>
> We thank the reviewers again for their constructive comments! We will add the clarifications mentioned in the comments and responses in the revision.
>
> A. Why is the energy function for faithfulness computed with respect to the reference summary and not the source articles? As also stated by the authors, faithfulness is associated with generating text that is inferable from the source document.
>
> We did try using the source article as the reference for faithfulness in our experiments but we found the computation efficiency is extremely low. Therefore, we use the reference summary as an approximation in the final experiments.  From the experimental results, we found that the faithfulness calculated with reference summary can still improve the factual consistency.
>
> B. What is the intuition behind using the similarity-based NCE loss in equation 1?
>
> It is found that similarity-based contrastive learning objective is efficient in training the discriminator. Therefore we follow CLIFF on using this loss.
>
> C. Some of the parameters in the equations are not described such as P and N.
>
> P is the positive sample set and N is the negative sample set. Thank you for raising this question. We will add the clarification to the revision.
>
> D. The results show (and the authors emphasize) that self-supportiveness is not a challenging problem for the current systems. Do you think the results would change if you remove the self-supportiveness loss term?
>
> According to the experiments, we believe it will not change greatly.
>
> E. Can energy score be applied while decoding with sampling and not beam decoding?
>
> It is applicable to any sampling since the energy scorers take the decoded sentence(s) as the input.
>
> F. Xsum consists of one-line summaries. What role do you think the length of the summaries can play when optimizing for the proposed energy functions?
>
> Thanks for the insightful question. Our method is not sensitive to the summary length. However, given the fact that short summaries have fewer factual consistency issues, in the beam search we can consider reranking the candidates by considering both the energy scores and generation probability scores, for example, assigning the energy score a smaller weight for shorter summaries. We will leave it for future work.

---

### Meta-Review · Area_Chair_3i8L · 2023-09-14

**Recommendation:** 3

**Metareview:**

In order to assess the accuracy, factualness, and coherence of summaries, this paper defines three energy functions embedded into a supervised summarization model. To enhance the corresponding characteristics of the system-generated summaries, the energy functions are included in both the training and inference processes. The generated sequences are reranked according to these scores during the inference stage.

Pros:
1. The proposed integration of consistency-based energy functions into summarization models is definitely a novel approach;
2. The paper is well-written;
3. The evaluation results are clearly described.

Cons:
Accuracy performance which is comparable to SOTA baseline systems with multiple metrics, but not always outperforming.

The authors provided very detailed answers to all reviewers. Some issues that were raised by reviewers are left for future work.

---

### Decision · Program_Chairs · 2023-10-07

**Decision:**

Accept-Findings

**Comment:**

In order to assess the accuracy, factualness, and coherence of summaries, this paper defines three energy functions embedded into a supervised summarization model. To enhance the corresponding characteristics of the system-generated summaries, the energy functions are included in both the training and inference processes. The generated sequences are reranked according to these scores during the inference stage.

Pros:
1. The proposed integration of consistency-based energy functions into summarization models is definitely a novel approach;
2. The paper is well-written;
3. The evaluation results are clearly described.

Cons:
Accuracy performance which is comparable to SOTA baseline systems with multiple metrics, but not always outperforming.

The authors provided very detailed answers to all reviewers. Some issues that were raised by reviewers are left for future work.